# Environmental and Economic Life-Cycle Assessments of Household Food Waste Management Systems: A Comparative Review of Methodology and Research Progress

**Na Yang** [1,2], **Fangling Li** [1,2], **Yang Liu** [1,2], **Tao Dai** [1], **Qiao Wang** [1], **Jiebao Zhang** [3], **Zhiguang Dai** [1] **and Boping Yu** [1,2,*] 

1   Shenzhen Academy of Environmental Sciences, 50 Honggui Road, Shenzhen 518001, China;
    cjl-1907@163.com (N.Y.); fanglingli_2007@163.com (F.L.); yangtze_as@163.com (Y.L.); cndaitao@163.com (T.D.);
    wang_qiao_wq@163.com (Q.W.); szdai168@vip.163.com (Z.D.)
2   Guangdong Environmental Protection Sewage High Quality Utilization Engineering Technology R&D Center,
    Shenzhen 518001, China
3   Shenzhen Municipal Solid Waste Sorting Management Service Center, Shenzhen 518001, China;
    zhangjb220505@163.com
*   Correspondence: xiaobo529@126.com

**Abstract:** Household food waste (HFW) is the main component of municipal solid waste (MSW). Appropriate HFW management strategies could reduce the environmental burdens and economic costs to society. Life-cycle thinking is an effective decision-making tool for MSW management. This paper compares the three main environmental and economic assessment methodologies, i.e., societal life-cycle costing (societal LCC), environmental cost-effectiveness (ECE) analysis, and multicriteria analysis (MCA) in terms of the definitions, method frameworks, and their advantages/disadvantages. Most reviewed studies applied the environmental life-cycle costing (ELCC) method, a simplified ECE, which does not involve interactive quantitative comparisons between environmental and economic benefits. Further attention should be paid to the coordination between life-cycle assessment (LCA) and life-cycle costing (LCC), the monetization coefficient in external cost calculation of societal LCC, and the standardization and evaluation approaches of ECE. HFW prevention is rarely considered in the reviewed literature but was demonstrated as the best route over treatment or utilization. Anaerobic digestion is environmentally preferable to composting and landfilling; it is comparable to biodiesel production, feeding conversation, and incineration. From the perspective of economic costs (including societal LCC), the ranking of treatment technologies varied a lot from one study to another, attributable to the diverse evaluation methods and different data sources. To improve the environmental and economic assessment approaches to HFW management, an inventory database (e.g., food waste properties, technical treatment parameters, material flow, and fund flow data) suitable for HFW should be constructed. When establishing the system boundaries, the processes of source sorting, collection and transportation, and by-product handling should be coherent with the investigated treatment technology.

**Keywords:** waste classification; source sorting; life-cycle costing; life-cycle assessment; inventory; anaerobic digestion

## 1. Introduction

Along with rapid urbanization, huge amounts of food waste (FW) are generated worldwide. FAO [1] estimated that approximately 27% of food produced is wasted annually. FW management hierarchy has been recommended by the European Commission [2] as prevention, re-use, recycling, recovery, and treatment. According to its generated sources, such as households, restaurants, markets, food industries, etc. [3], FW's characteristics and management routes differ. For example, FW generated in restaurants is mainly food scraps

that are usually cooked. Nevertheless, FW generated from households includes inedible parts of vegetables, meats, and fruit, usually uncooked. FWs generated from markets and food industries are easy to collect separately because they are centralized generated. To eliminate the production of illegal cooking oil, FW generated in restaurants and canteens has been independently collected and treated in more than 100 Chinese cities since 2010 [4]. In contrast, FW generated from households, defined as household food waste (HFW), is usually mixed and collected with other municipal solid waste (MSW) components in many countries and regions [5], resulting in vast environmental impacts [6]. HFW, rather than other FW, is the research objective in this study.

HFW is the main component of MSW, e.g., accounting for >50% of the total MSW in wet weight [7–10] in developing countries. HFW has high moisture content and rapid biodegradation, causing several disadvantages to the MSW management system. For example, HFW sticks to other waste components, decreasing the value of the recyclables. The high content of HFW would reduce the lower heating value (LHV) of MSW and subsequently the energy recovery efficiency of MSW incineration, which induces higher GHG emissions. Moreover, the degradation of HFW in landfill sites and incineration plants leads to secondary pollution, such as leachate and odor [11]. Therefore, source separation of HFW and appropriate treatment approaches for resource recovery would reduce the environmental impacts of the MSW management, thereby promoting the achievement of carbon neutrality goals [12,13]. Several European countries have implemented HFW sorting as a national policy [14]. Since 2019, China has promoted zero-waste city construction and waste classification. HFW separation from MSW for individual treatment is one of the main policy goals. Up to now, various management strategies of HFW exist in different regions. For HFW mixed with other MSW, landfilling or incineration is usually the treatment method [15,16]. For source-separated HFW from MSW, anaerobic digestion and aerobic composting are commonly used for energy or resource recovery [17]. In the United States, food waste disposers (FWDs) are introduced to some households, by which HFW is ground in the kitchen and discarded through the sewer for treatment with wastewater [11]. In some Chinese cities, HFW is pre-treated by high-pressure extrusion and delivered to incineration plants [18]. It remains inconclusive to identify an optimal HFW management strategy according to the local waste characteristics, production amount, residential type, infrastructure construction, and energy structure [16,19,20].

HFW management system, as municipal infrastructure, aims to decrease the environmental burdens for the whole society. Establishing and maintaining an HFW management system requires large amounts of economic investment. In addition, appropriate HFW utilization could recover energy or materials, thereby creating financial revenues. Therefore, environmental and economic performance should be assessed when identifying the optimal HFW management strategy. Life-cycle approaches are highly effective decision-making tools that quantify, evaluate, and compare various products, services, and technologies. In recent years, they have been widely used in MSW management [9,21,22]. Life-cycle assessment (LCA) usually evaluates the environmental impacts of HFW management. The system boundaries, methodologies, and input data varied between the LCA studies, leading to different comparisons between the alternatives [23]. A review of 19 studies for assessing GHG emissions from HFW treatment scenarios found that the selection of energy- and/or bio-system substitution was highly relevant to the results [24]. LCC has been applied and integrated with LCA for comprehensive economic and environmental assessments. De Menna et al. [19] pointed out that LCC methodologies are still immature due to the inconsistent principles between LCC and LCA. For this reason, the results for the preferred scenarios in the reviewed studies are not discussed. In addition, most of the FW taken into account by De Menna et al. [19] was generated from canteens and food production chains, while only two studies [25,26] dealt with FW generated from households coherently with this study's research objective.

The primary purpose of this study is to conduct a systematic literature review on the definitions, differences, and approaches of the main environmental and economic assess-

ment methodologies. Then, the research progress of those methods on HFW management is investigated to identify the preferable strategies. Ultimately, the defects of the existing studies are identified, and research recommendations are made to promote using environmental and economic assessment methods in decision making for HFW management.

## 2. Environmental and Economic Assessment Methods

Environmental and economic assessment refers to a comprehensive quantitative analysis of the environmental impacts and economic costs of public policies or projects to answer the question, what policy or project is better to achieve lower environmental impacts and affordable economic costs? The results would be adopted to guide the effective allocation of social resources and improve government decision-making quality. The comprehensive quantitative analysis of environmental and economic benefits has not yet formed a unified theoretical system. Commonly used methods mainly include environmental cost–benefit analysis (E-CBA), eco-efficiency analysis, and MCA. The comparison among the three methods is shown in Figure 1.

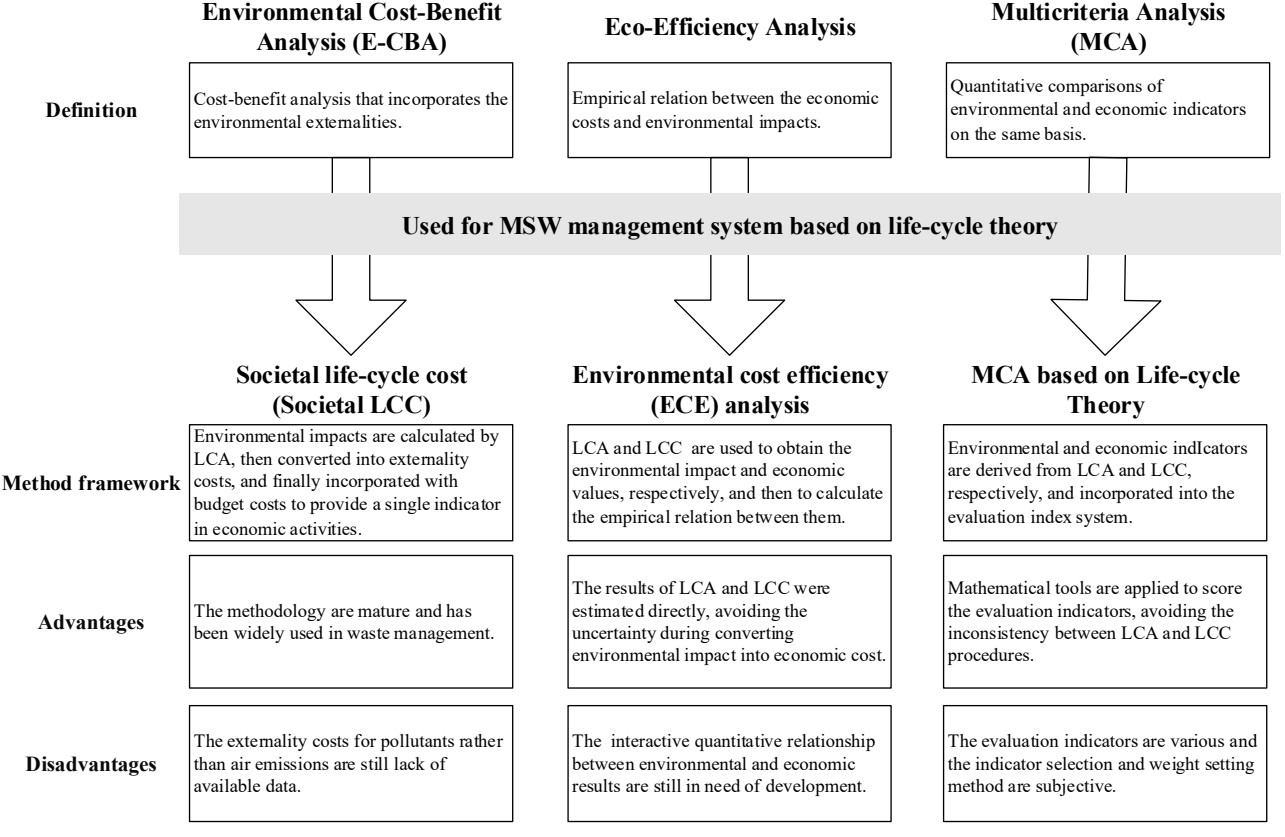

**Figure 1.** Comparison among the environmental and economic assessment methods.

### 2.1. Environmental Cost–Benefit Analysis (E-CBA)

E-CBA refers to cost–benefit analysis (CBA) that incorporates the environmental externalities by assigning corresponding economic values to environmental impacts or benefits and maximizing the net economic benefits [27,28]. E-CBA is crucial in environmental policymaking, such as the Clean Air Act of the United States and the Landfill Directive (1999/31/EC) in the European Union [28–31].

Performing E-CBA mainly includes three steps. First, the input and output of the policy or project are identified according to the research object, the temporal scale, and geographical range. During this stage, the environmental impacts should be evaluated, i.e., monetizing environmental impacts by establishing relationships between the environmental pollutant emissions or environmental quality changes and the economic costs. The positive and negative environmental impacts of the policy or project are analyzed. Positive

environmental impacts (i.e., environmental benefits) and economic benefits constitute the total benefits, whereas negative environmental impacts (i.e., environmental damage) and economic costs constitute the total cost. Second, various costs and benefits are temporally normalized using a discount rate. Third, the total costs and benefits are compared through evaluation methods, such as the net present value and cost-effectiveness ratio methods. The key process to CBA is environmental valuation, which aims to reveal the impact of changes in environmental quality on human welfare. Evaluation methods include direct market valuation, revealed preference, and survey evaluation [27,32].

## 2.2. Eco-Efficiency Analysis

Eco-efficiency indicates the empirical relation between the economic costs and environmental impacts of a project. According to specific empirical relationships, there are four main types of eco-efficiency, i.e., environmental productivity and its inverse, the environmental intensity of production, environmental improvement cost and its inverse, and environmental cost effectiveness [33]. As one of the instruments of industrial ecology, eco-efficiency is widely used at the enterprise level to promote product design and manufacturing system development. For instance, BASF Group has developed "Eco-efficiency Management" application software, which evaluates the comprehensive benefits of the enterprise in terms of raw material consumption, energy consumption, land use, and pollutant discharge. The results can help with strategic decision making and product development [34].

The analytic framework for eco-efficiency includes two steps. First, the economic and environmental parts are separately quantified into single scores. Second, these two scores are combined into the desired eco-efficiency ratio. In practice, the eco-efficiency of a single project is rarely useful, while this method is generally used to identify the optimal eco-efficiency ratio of multiple projects. Eco-efficiency is calculated in several ways: (1) The standardization method directly calculates the ratio of economic and environmental scores after data standardization. Kicherer et al. [35] transformed economic and environmental indicators into dimensionless scores. The calculation is relatively simple, but the results are diverse, with poor comparability between projects. (2) The optimal curve method drops the standardized economic and environmental scores in a two-dimensional coordinate graph and draws the potential optimal state curve among multiple projects [33]. The optimal state point is then selected based on economic and environmental weight trade-offs. (3) Data envelopment analysis (DEA) compares multi-input and multi-output indicators without standardization and weighting. DEA has been used in regional-scale [36] and industry [37] eco-efficiency analyses. In general, the eco-efficiency method avoids the uncertainty of the environmental evaluation process in the E-CBA method. The weakness of eco-efficiency is the lack of a unifying methodology to combine economic and environmental benefits [38].

## 2.3. Multicriteria Analysis (MCA)

MCA is a method that comprehensively considers multiple or even conflicting target criteria to make decisions in limited or unlimited projects [39]. MCA consists of three steps. First, representative indicators are selected to form an evaluation index system. The environmental and economic indicators should be covered if used for environmental and economic assessment. Second, the evaluation criteria and weights of each index are determined. Third, the mathematical model is applied to comprehensively calculate the evaluation score of each project, identifying the optimal one. This method can achieve quantitative comparisons of various criteria on the same basis and serve in energy planning [40] and environmental policymaking [41,42]. However, it is highly subjective on the index selection, weight setting, and evaluation score, increasing the uncertainty of the results. Specific calculation methods include the weighted sum method [43], analytical hierarchy procedure (AHP) [44], fuzzy mathematical model, and technique for order preference by similarity to an ideal solution (TOPSIS) [45].

### 3. Environmental and Economic Assessment of MSW Management Based on Life-Cycle Theory

Conventional environmental and economic assessment is mainly used in construction projects or industrial products and services. The primary objective of the MSW management system is to decrease environmental pollution rather than create economic benefits [46]. During the environmental and economic assessment of construction and industrial processes, the current costs and benefits are usually considered. In contrast, the assessment approach for the MSW management system considers the environmental impacts of its whole life cycle in addition. LCA and life-cycle costing (LCC) have been integrated to form the following three environmental and economic assessment methods [47,48]: societal life-cycle cost (societal LCC) analysis based on E-CBA, environmental cost efficiency (ECE) analysis based on the eco-efficiency method, and MCA.

#### 3.1. Societal LCC

Societal LCC analysis converts environmental impacts into externality costs by monetary evaluation. It then incorporates them with budget costs to estimate the overall societal cost of the MSW system, also called life-cycle cost–benefit analysis (LC-CBA) [49]. Systematic methodologies have been established for societal LCC in waste management, e.g., a Nordic guideline for cost–benefit analysis in waste management. The monetary evaluations are used to monetize the impacts of pollutant emissions on health, environment, and entertainment, which is a key parameter in societal LCC. The accounting prices of pollutant emissions are often used for monetary evaluations. A single type of pollutant emission amount and its accounting prices were individually multiplied and summarized to obtain the externality costs of environmental impacts [50]. A monetizable pollutant inventories and accounting price databases for air emissions were proposed by Martinez-Sanchez et al. [50]. Opportunity costs of land and disamenity were considered to represent the externality cost of waste treatment plants [49]. The accounting prices of water emissions could be derived from the cost of pollutant abatement [20]. However, in comparison with the accounting prices of air emissions, which use impact pathway analysis, dose–response relationship, and "willingness to pay" method to estimate the damage cost of a specific pollutant, the accounting prices of water emissions are still immature and need more research. Presently, the research on monetary evaluations is mainly from Europe and the USA. As the economic performance of environmental impact differs from one place to another, monetary evaluations should be best derived from the locally available literature. In addition, transparency of the estimation process and applicability to a specific study should also be considered when selecting monetary evaluations [50]. In summary, the most notable externality costs for MSW management systems are air emissions. However, the externality costs of water and soil pollution, resource consumption, and social impacts (such as time cost, odor, noise, landscape, and traffic congestion) have rarely been implemented in the calculation framework.

#### 3.2. Environmental Cost Efficiency (ECE)

To calculate the ECE, LCA and LCC were used to obtain the MSW management's environmental impact and economic values, respectively. Then, the empirical relation between them was calculated [46]. There are three eco-efficiency application options [33]: (1) incremental eco-efficiency (E/Eincr), which represents the total economic value and its total concomitant environmental effects in a specific scenario; (2) win-win eco-efficiency (E/Ewinwin), which compares the environmental burden and economic costs between the potential improvement scenario and a historical reference situation; and (3) paired eco-efficiency (E/Epairwise), which compares environmental burden and economic costs between any two potential improvement scenarios. In the studies for the MSW management system (Table 1), the definition of ECEs remains diverse. Correspondingly, the normalization method between environmental and economic values and the evaluation method among different scenarios is crucial. Yang et al. [51] used GDP per capita and $CO_2$-equivalent

emissions per capita to normalize the environmental and economic values, respectively, with the unit after normalization as "person·yr". The normalized environmental–economic value ratio was calculated to represent the ECE for each optimization measure. The ECEs for various measures were compared directly with each other. Hellweg et al. [46] pointed out that the paired eco-efficiency is meaningful for the end of pipe treatment technologies, such as the MSW management system. In a 3E + S model (3E denotes environment, energy, and economics, while S denotes society) [52], the ratio of environmental indicators to economic indicators was directly calculated to represent the ECE with no normalization. Then, the ECEs of the two proposed technologies were displayed in a data matrix for comparison. Huppes and Ishikawa [26] recommended finding the trend of system eco-efficiency optimization through the "optimum envelope" method rather than normalization because the latter can cause inconsistent results and incomparability between cases. Elsewhere, normalization and "optimal envelope" methods could also be integrated by drawing environmental and economic values of the investigated scenarios in a two-dimensional graph [38,53]. In general, developing the ECE method requires interactive quantitative analysis of environmental and economic results to achieve synergy between LCA and LCC.

**Table 1.** Approaches used for environmental cost efficiency in reviewed studies for the MSW management system.

| Literature | Categories | Definitions | Normalization Methods | | Units | Evaluation Methods |
|---|---|---|---|---|---|---|
| Mah et al. [54] | $E/E_{INCR}$ | The effects of the total concomitant environmental impacts and its economic cost. | Env. [a] | No normalization. | $kgCO_2$-eq·$t^{-1}$ | Draw the environmental impacts and economic costs of the investigated scenarios in a scatter plot. |
| | | | Eco. [a] | No normalization. | MYR·$t^{-1}$ | |
| Yang et al. [51] | $E_{WIN-WIN}$ | The ratio between the environmental improvements of the optimization measure compared with the current situation and the economic costs of the optimization measure. | Env. | Normalized by using per capita environmental impact in east China. | person·yr | Calculate the ratio of normalized environmental indicators and economic indicators. |
| | | | Eco. | Normalized by using GDP per capita. | person·yr | |
| Woon and Lo [55] | $E/E_{PAIRWISE}$ | The relative impact of the economic aspect on the ecological destruction of the proposed situations. | Env. | Normalized by calculating the relative change in percentage of environmental impacts for a specific situation to the reference one. | % | Draw the environmental impact and economic cost using a two-dimensional graph. Then, compare the variation trend of different situations. |
| | | | Eco. | Normalized by calculating the relative change in percentage of economic costs for a specific situation to the reference one. | % | |

**Table 1.** *Cont.*

| Literature | Categories | Definitions | | Normalization Methods | Units | Evaluation Methods |
|---|---|---|---|---|---|---|
| Hellweg et al. [46] Ren and Yang [52] | | The environmental benefit of a technology A over a technology B per additional cost. | Env. | No normalization. | eco-indicator points·t$^{-1}$ | The ECEs of the two proposed technologies were displayed in a data matrix. The data represent the environmental advantage per monetary unit of the technology in the column over the technology in the line. |
| | | | Eco. | No normalization. | Euro·t$^{-1}$ | |
| Zhao [38] | | The economic value and its concomitant environmental burden between two alternative technologies. | Env. | Normalized by using per capita environmental impacts. | person·yr | Draw the eco-efficiencies of the alternative technologies in scatter plots with environmental burden and economic value as X and Y axes. The lines joining any two plots are transformed as a potential optimum envelope. The optimal alternative technology on the envelope depends on the trade-off theory. |
| | | | Eco. | Normalized by using GDP per capita in the baseline year. | person·yr | |

[a] Abbreviations: Env.: Environmental value; Eco.: Economic value.

## 3.3. Multicriteria Analysis (MCA) Based on Life-Cycle Theory

For MCA based on life-cycle theory, environmental and economic indicators are evaluated from LCA and LCC, respectively, and incorporated into the evaluation index system (Table 2). As the indicators are in different units, normalization or standardization is necessary to make the indicators comparable to each other. Then, the weights of individual indicators are determined to judge the relative importance of each one. The values of each indicator are intergraded to a comprehensive score for a specific scenario using a mathematical method. Finally, the scenarios are ranked to identify the best ones by maximizing economic benefit and minimizing environmental impacts. In an environment–energy–economic (3E) assessment model for the MSW management systems [47], environmental burden and energy consumption were evaluated by LCA procedures, while economic performance was obtained through LCC. AHP was used to determine the indicator's weight, which could quantify experts' empirical judgment. TOPSIS was used for the final ranking of the evaluated scenarios. A fuzzy mathematical evaluation model [45] could also be conducted to evaluate FW anaerobic digestion technology. The environmental–economic benefits were divided into five grades, and the standard index values for each evaluation grade referred to experimental and literature data. Vinyes et al. [56] transformed the evaluation indicators into contribution percentages and obtained three sustainability factors (SFs) for environmental, economic, and social dimensions after the standardization and normalization of the indicators. The SFs of different dimensions are evaluated by qualitative description individually but lack comprehensive quantitative comparison among each other. In short, the MCA is to apply mathematical tools to score the evaluation indicators obtained by LCA and LCC procedures, which could avoid the inconsistency between those two methods. However, MCA is rarely used in MSW management, whereas previous studies had various evaluation indicators and were subjective in indicator selection and weight setting.

**Table 2.** Approaches used for multicriteria analysis method in reviewed studies for the MSW management system.

| Literature | Indicators | | | Weighting Methods | Indicator Evaluation Methods | Comprehensive Evaluation Methods |
|---|---|---|---|---|---|---|
| | Environment | Economic | Others | | | |
| Chen et al. [45] | GWP; FETP; HTP; AP; EP. | Cost; Benefit; The ratio of profit to cost. | **Energy consumption**: Net energy input; Net energy output; Energy recycling rate. | Experts Grading and AHP. | The indicator values are divided into five grades: very good, good, average, bad, and very bad, which are expressed as 5, 4, 3, 2, and 1 scores. The standard indicator values referred to experimental and literature data. | The comprehensive score is calculated based on the values and weights of each indicator. |
| Dong et al. [47] | Human health; Ecosystem quality; Resources. | Investment cost; Operation cost; Avoided cost. | **Energy**: Fuel consumption; Electricity consumption and recovery; Fuel production; Auxiliary materials production. | AHP. | The environmental factor is represented by weighted environmental impact with the unit of "Pt (one person per year)". The economic factor is represented by net LCC cost with the unit of "CNY·t$^{-1}$". The energy factor is represented by the net energy consumption with the unit of "MJ·t$^{-1}$". | TOPSIS matrix. |
| Vinyes et al. [53] | ADP; AP; EP; GWP; ODP; HTP; FETP; MAET; TET; POP; Energy consumption. | Economic cost | **Social:** Employee education level; Equal opportunities; Environmental education; Local employment; Public commitments to sustainability issues; Contribution to economic development. | No weighting. | The indicators are transformed into contribution percentages by comparing the alternative scenarios and then scored from 1 to 5. Each individual sustainability factor (SF), as $SF_{environment}$, $SF_{economy}$, $SF_{social}$, is calculated by summing the indicators of its dimension and then recalculated into relative values (between 0 and 1). | Qualitative description for individual SF. |

## 3.4. Coordination between LCA and LCC

The comprehensive environmental and economic assessment method based on life-cycle theory for MSW management is still immature. LCA and LCC derived from environmental and economic research perspectives, respectively, differ in theoretical principle and method design. When researchers integrate the results of LCA and LCC, they generally encounter the coordination of the two theoretical systems on system boundaries, distribution, and discounting (Table 3).

**Table 3.** Uniformity between LCA and LCC methods in the reviewed studies.

| Literature | System Boundary | | Allocation Method | Discount Rate |
|---|---|---|---|---|
| | Similarity | Difference between LCA and LCC | | |
| Martinez-Sanchez et al. [25] | Source separation, collection, transportation, treatment, and disposal. | Only budget costs are considered in conventional LCC. Externality costs are converted from LCA. | Substitution is conducted for LCA based on material and energy recovery. | In total, 4% for LCC, as well as for the external costs calculated based on LCA. |
| Zhao et al. [53] | Collection, transportation, treatment, by-product utilization, and residue disposal. | Plant construction and decommissioning are ignored in LCA, yet it is calculated in LCC. | Economic partitioning is conducted for both LCC and LCA. The allocation factors are created according to their market price. | Not mentioned. |
| Ren and Yang [52] | Collection, transportation, and end-of-pipe treatment. | LCC calculates the design cost, the opportunity cost of land, and disamenity due to treatment plant construction. | Substitution is conducted for both LCC and LCA based on electricity production. | Not mentioned. |
| Yang et al. [51] | Collection, transportation, and end-of-pipe treatment. | Constructing the treatment plant was excluded in LCA, yet it is considered in LCC. | Substitution is conducted for LCC and LCA based on electricity generation and fertilizer utilization. | Not mentioned. |
| Dong et al. [47] | MSW treatment, leachate treatment, electricity generation. Collection and transportation are excluded as they are identical in all scenarios. | Plant construction and decommissioning are ignored in LCA, yet it is calculated in LCC. | Based on electricity production, substitution with system expansion is chosen for LCC and LCA. | In total, 5% for LCC. |

**(1) System boundary.** Both LCA and LCC methods follow the "cradle-to-grave" principle and include all functional unit-related processes as much as possible. LCA generally excludes the environmental impacts of treatment facility construction and demolition process, whereas LCC usually consists of the economic costs of treatment facility design and construction.

**(2) Allocation.** MSW management system is a multi-output process with MSW management service products as the primary product, while the production of biogas, electricity, fertilizer, etc., are by-products. Thus, the input material flows, energy flows, and pollutant emissions must be partitioned between the MSW management service and the co-products. There are two major approaches to implementing the allocation solution. Most studies use the substitution method (or extended systems method). The environmental impacts or

economic costs of co-products generated by the "normal" process are subtracted from the MSW management system. In contrast, several researchers use the partitioning method, such as the economic value allocation method [52,53,57]. Here, the MSW management system is split into several independent processes, with the allocation parameters determined according to the market value of each co-product. Zhao [38] compared the two allocation methods and found that the results obtained by the substitution method were much smaller than those obtained by the economic value allocation method. The substitution method's problem is selecting the appropriate avoided process, and the amount of inventory data required is relatively large due to the expansion of the system boundary. The disadvantage of the economic value allocation method is that the market value of the co-products fluctuates over time.

**(3) Discounting.** Economic value usually needs to be discounted to present value in an economic analysis method. Currently, there is no widely accepted consistent discount rate. Generally, the environmental analysis does not need discounting, considering that the pollutant emissions which occurred at different times induce the same impact on the environment. The studies using the societal LCC method generally convert environmental impacts into economic costs and then discount the economic cost into present value [20,25,50,54,55]. There are various discount rates, such as the instant interest rate, social discount rate suggested by local authorities, or discount rate for infrastructure projects. Therefore, the discount rate varies widely from 1.2% to 4%. Studies using the ECE method are generally not discounted. For instance, Zhao [38] suggested that the duration of the MSW treatment process does not significantly impact the economic cost, thereby ignoring cost discounting. For some of the MCA studies, the LCC results were also discounted to present value [47].

## 4. Research Progress of Environmental and Economic Life-Cycle Assessment of HFW Management

The objective of this study is to offer a systematic review of the research progress of environmental and economic assessment of HFW management. Due to the lack of consistent terminology for the food waste generated from households, related keywords such as "household food waste", "food waste", "household kitchen waste", "organic waste", "organic fraction of municipal solid waste", plus "environmental", and "economic" were used to conduct a topic search in the Web of Science database. Then, the abstracts and key contexts were carefully reviewed to eliminate studies referring to food waste that is not generated from households, such as waste from restaurants, markets, the food industry, etc. Consequently, 12 studies were screened out. The scope and goals, methodologies, main results and data sources of the remaining papers are shown in Tables 4–6.

**Table 4.** Information from the reviewed studies on the environmental and economic life-cycle assessment of HFW management (scope and goals).

| Literature | Functional Units [a] | Scope and Goals | | | | | |
|---|---|---|---|---|---|---|---|
| | | System Boundaries | | | | | |
| | | C&T [b] | Pre-treatment | Treatment | By-Product Handling | Others | Expansion [c] |
| Kim et al. [26] | 1 tonne of FW | √ [d] | / [e] | √ | √ | / | No |
| Carlsson et al. [58] | 1 tonne of source-sorted FW | / | √ | √ | √ | / | No |
| Martinez-Sanchez et al. [25] | FW generated in 1 year | √ | / | √ | / | Food production | Residual MSW |
| Eriksson et al. [59] | FW generated in 1 year | √ | / | √ | √ | Source separation or central sorting | Residual waste, sewage sludge |
| Ahamed et al. [60] | 1 tonne of FW | √ | / | √ | √ | / | No |

**Table 4.** *Cont.*

| Literature | Scope and Goals | | | | | | |
|---|---|---|---|---|---|---|---|
| | Functional Units [a] | System Boundaries | | | | | |
| | | C&T [b] | Pre-treatment | Treatment | By-Product Handling | Others | Expansion [c] |
| Maalouf and El-Fadel [8] | FW generated in 1 year | √ | / | √ | √ | Material fraction recycling | Remaining waste, WW, sewage sludge |
| Bong et al. [7] | 1 tonne of OW | √ | / | √ | √ | / | Oil palm fresh fruit bunch |
| Edwards et al. [20] | FW generated in 1 year | √ | / | √ | / | / | Inert waste, garden waste, sewage sludge |
| Slorach et al. [16] | 1 tonne of FW | √ | | √ | / | Treatment plant construction | No |
| Mayer et al. [14] | 1 kWh of exergy or 1 kg of OFMSW | √ | √ | √ | √ | / | No |
| Yu and Li [61] | 1 tonne of MSW | √ | / | √ | √ | Source separation | Residual waste |
| Yong et al. [62] | 50% of OFMSW generated in Malaysia | / | √ | √ | √ | / | No |

[a] To be consistent with the original studies, the terms for household food waste used in the literature are shown here. [b] C&T, collection and transportation. [c] All of the reviewed studies considering by-product handling process expanded the system boundary to include the substituted commercial products. This is considered as a kind of allocation method, but not described in this table. [d] "√", this process was included in the system boundary of the reviewed literature. [e] "/", this process was not included in the system boundary of the reviewed literature.

**Table 5.** Information from the reviewed studies on the environmental and economic life-cycle assessment of HFW management (methodology and results).

| Literature | Methodology | Results (Scenario [a] Ranking [b]) | | | | | | | |
|---|---|---|---|---|---|---|---|---|---|
| | | Benefits [c] | AD | COMP | FD | INC | LF | FWD | Others |
| Kim et al. [26] | SLCC: Benefit–cost ratio | Total | 4 | 3 | 1 | 2 | 5 | √ | / |
| Carlsson et al. [58] | ELCC: Qualitative comparison | Env./Eco. | √ | / | / | / | / | / | Increasing TS concentration in AD: 1 Increasing TS distribution to AD: 2 Decreasing electricity consumption: 3 |
| Martinez-Sanchez et al. [25] | ELCC: Qualitative comparison | Env. | 2 | / | 2 | 2 | / | / | Prevention of edible FW:1 |
| | ELCC: Qualitative comparison | Eco. | 3 | / | 2 | 4 | / | / | Prevention of edible FW: 1 |
| | SLCC: Absolute costs | Total | 4 | / | 3 | 2 | / | / | Prevention of edible FW: 1 |
| Eriksson et al. [59] | ELCC: Qualitative comparison | Env. | √ | / | / | √ | / | / | Central sorting: 1 Source separation: 1 |
| | ELCC: Qualitative comparison | Eco. | √ | / | / | √ | / | / | Central sorting: 1 Source separation: 1 |

**Table 5.** *Cont.*

| Literature | Methodology | Results (Scenario [a] Ranking [b]) | | | | | | | |
|---|---|---|---|---|---|---|---|---|---|
| | | Benefits [c] | AD | COMP | FD | INC | LF | FWD | Others |
| Ahamed et al. [60] | ELCC: Qualitative comparison | Env./Eco. | 1 | / | / | 3 | | / | FWEB: 2 |
| Maalouf and El-Fadel [8] | SLCC: Absolute costs | Total | 2 | 3 | / | / | 4 | 1 | / |
| Bong et al. [7] | ELCC: Qualitative comparison | Env./Eco. | / | 1 | / | / | 2 | / | Small-scale composting: 3 |
| Edwards et al. [20] | ELCC: Summing up of budget costs and transfer costs | Total | 6 | 2 | / | / | / | 5 | Household composting: 1 Co-digestion:4; MBT:3 |
| | SLCC: Summing up of budget costs and externality costs | Total | 6 | 2 | / | / | / | 5 | Household composting: 1 Co-digestion:3; MBT:4 |
| Slorach et al. [16] | Ranking and score | Env. | 1 | 4 | / | 2 | 3 | / | / |
| | | Eco. | 3 | 2 | / | 1 | 4 | / | / |
| | | Total | 2 | 3 | / | 1 | 4 | / | / |
| Mayer et al. [14] | ELCC: Qualitative comparison | Env./Eco. | 1 | / | / | 2 | / | / | Pre-drying prior to INC:3; AD + solid digestate INC: 4 |
| Yu and Li [61] | SLCC: Summing up of environmental costs, household time costs, and internal costs | Env. | 2 | / | / | 1 | / | / | / |
| | | Eco. | 1 | / | / | 2 | / | / | / |
| | | Total | 2 | / | / | 1 | / | / | / |
| Yong et al. [62] | ELCC: Qualitative comparison | Env./Eco | 1 | / | / | / | 2 | / | / |

[a] Scenario abbreviations: AD: anaerobic digestion; COMP: centralized composting; FD: feeding; INC: incineration; LF: landfilling; FWD: food waste disposer; FWEB: food waste-to-energy biodiesel; MBT, mechanical biological treatment. [b] The lower the ranking number, the better the benefits. [c] Benefit abbreviations: Env.: Environmental benefits; Eco.: Economic benefits.

**Table 6.** Information from the reviewed studies on the environmental and economic life-cycle assessment of HFW management (data sources).

| Literature | Data Sources | | | | | |
|---|---|---|---|---|---|---|
| | Local Survey | Market Price | Modelling | Experimental | Literature | Database |
| Kim et al. [26] | Amounts and characteristics of FW. Process data for individual treatment stage. | Carbon trading price. | / | / | / | / |
| Carlsson et al. [58] | Energy use and generation, costs for pre-treatment facility. | Local cost of petrol and diesel. Market prices of N, P, and K in fertilizers. | / | Composition and methane potential of FW. | GHG emissions due to energy use. | / |
| Martinez-Sanchez et al. [25] | / | / | / | / | Inventory data. Accounting prices for pollutant emissions. | Ecoinvent database for LCA. |

**Table 6.** *Cont.*

| Literature | Data Sources | | | | | |
|---|---|---|---|---|---|---|
| | **Local Survey** | **Market Price** | **Modelling** | **Experimental** | **Literature** | **Database** |
| Eriksson et al. [59] | Processes data for source separation, collection, and central sorting. Economic data by expert estimation. | / | Process data for digestate treatment. | / | Process data for digestate treatment. | / |
| Ahamed et al. [60] | Inventory data for incineration. | / | / | Inventory data for biodiesel production and AD. | / | CED database for energy consumption and production. |
| Maalouf and El-Fadel [8] | / | / | | / | Environmental cost and saving. Economic data. | / |
| Bong et al. [7] | Economic costs. | / | / | / | GHG emission estimation. | / |
| Edwards et al. [20] | Monetary evaluation for water emissions. | / | / | / | Economic data. Monetary evaluation for air emissions. | / |
| Slorach et al. [16] | Inventory data for environmental impacts of FW treatment plants. | / | / | / | Composition of FW. Inventory data for environmental impacts. Economic data. | Ecoinvent database for LCA. |
| Mayer et al. [14] | / | / | Process data for incineration. | Composition and Biomethane potential of OFMSW. Pollutant emissions during the intermediate storage of OFMSW. | Technical and operation parameters. | Ecoinvent database for LCA. |
| Yu and Li [61] | Monetized time cost for source separation. Process data for MSW treatment. | Carbon trading price. Tax for acidic potential and energy consumption. | / | / | / | / |
| Yong et al. [62] | / | / | / | Biomethane potential of OFMSW. | Characterization of OFMSW. Expenses for anaerobic biogas power plant. | / |

## 4.1. Scope and Goals

The scope and goals of a life-cycle theory include functional units, system boundaries, and scenarios. There are two types of scope and goal settings for the reviewed literature. First, from the perspective of per unit of waste, the functional unit is generally 1 ton of HFW [7,16,26,58,60,61], and the amount of HFW to produce 1 kWh of exergy [14]. The corresponding system boundary is generally the HFW management system from generation to the final disposal. The scenario settings are mostly used to compare the different HFW treatment technologies. Notably, as the mainstream technology for HFW treatment, anaerobic digestion has been considered in almost all the literature. Other

technologies include feed conversion, composting, biodiesel production, incineration, and landfilling. Second, from the perspective of whole city management, the functional unit is generally the HFW generated in a certain place for one year [8,20,25,26,59]. This approach enables the city-level design of HFW management scenarios, such as HFW source sorting and efficiency [25,61], ground HFW delivered into the sewage treatment system [8,20], and co-treatment of HFW with garden waste or sewage sludge [8,20,59]. This concept often involves co-treatment with other wastes, requiring the extension of the system boundaries to include co-treated waste management systems.

*4.2. Assessment Methodologies*

According to De Menna et al. [19] and Section 3 of this study, the assessment methodologies in the reviewed literature can be classified into three types. First, five papers used SLCC, or societal LCC defined in this study, to incorporate environmental benefits into the economic cost system [8,20,25,26,61]. The absolute costs [8,25,61] or benefit–cost ratios [26] were calculated to represent the comprehensive results. In a study by Kim et al. [26], landfilling is considered the worst scenario with the lowest benefit–cost ratio. However, landfilling is the second preferable scenario if the evaluating criteria are altered from benefit–cost ratios to absolute costs. This is due to the relatively lower budget costs and benefits of landfilling. Thus, selecting evaluating criteria should be paid special attention according to the data feature. Second, eight papers used environmental LCC (ELCC), similar to ECE method defined in this study [7,14,25,26,50,58–60,62]. However, these studies only qualitatively describe the relationship between environmental and economic benefits and do not involve interactive quantitative comparisons. Third, one paper adopted the ranking and score method, a simplified MCA method, which gives a combined score for environmental and economic results [16].

LCA is usually used for the calculation of environmental benefits but with different analysis steps in those three methodologies. SLCC uses LCA to calculate the amount of pollutant emissions. Then, it is converted into externality costs and incorporated into economic accounting, whereas ELCC and the ranking and score method calculate environmental impact potentials through a complete LCA. Furthermore, the types of environmental impacts investigated so far are diverse. For example, global warming potential, or greenhouse gas (GHG) emissions, was considered in all literature. Eriksson et al. [59] argued that GHG emissions should not be used as the only environmental impact category as carbon in HFW is of biological origin, imposing no GHG emissions when converted to carbon dioxide either by biological treatment or combustion. Acidification, eutrophication, photochemical oxidation, and energy consumption were also frequently investigated.

LCC or CBA methods are often employed to calculate the economic benefits but with different cost categories in those three methods. SLCC only calculates budgeted costs [8,20,25,26,61], including investment costs, collection, and transportation costs, operating costs for treatment plants, and revenues from by-product sales. The investment cost mainly refers to the amortization of fixed assets, while some researchers consider infrastructure construction [16,58,59,62] or land rentals [60,62]. The collection and transportation costs mainly refer to the fuel consumption [60,61]. The operating costs of treatment plants include feedstock consumption in terms of water, electricity, heat, chemicals, etc. [7,58,60]. Some studies [7,20,26] also considered labor costs using per-hour wage of the workers, and the working time. The workers refer to the lorry drivers and waste handlers rather than technical engineers. For the by-products, rather than the revenue from sales, researchers should pay attention to the economic cost during utilization, for example, transportation to the users [58], spreading on the land, and storage. In addition to calculating budget costs, ELCC and ranking and score methods also include transfers, such as taxes, waste disposal fees, biogas, or landfill gas power generation subsidies, representing the economic cost of environmental impacts [16,20,25,62].

The main data during environmental and economic assessments of HFW management include waste amounts, waste characteristics, material flow, and fund flow. For the existing

literature, the amounts and characteristics of HFW and the flow data during collection and transportation processes mostly arise from the local investigation. Moreover, the data related to the treatment process have various sources, such as the actual performance of the treatment plant [16,26,61], experimental data from small and medium-sized experiments [60], literature reports [8,25,60], model simulations [14], and expert estimates [59]. For the SLCC method, the monetary valuation of gas emissions was obtained from local research [20], literature reports [8,25], and carbon trading market prices [26,61]; the monetary valuation of water emissions means the pollution control costs per unit derived from government research reports [20].

### 4.3. Which Management Strategy Is Preferable?

Consistent with the FW management hierarchies, the prevention of edible FW is demonstrated as the best route over treatment or utilization [25]. This is due to the avoided food production process, which contributes to extremely high environmental and economic impacts in the scenarios without FW prevention. From the perspective of environmental benefit, anaerobic digestion is preferable to composting and landfilling and is comparable with biodiesel production, feeding conversation, and incineration [14,16,25,60–62]. When the economic costs are taken into account, the advantages of anaerobic digestion are not significant. Composting usually has lower economic costs than anaerobic digestion [16,20,26]. The ranking between anaerobic digestion and incineration varied a lot, owing to the data sources and treatment scales [14,26,60]. The inventory data of the anaerobic digestion process used by Ahamed et al. [60] came from a pilot project which limited the data reliability. In the study by Kim et al. [26], the higher economic costs of anaerobic digestion could be referring to the smaller scale and shorter lifespan of its treatment facilities, resulting in high construction and operating costs.

The upstream process before HFW treatment is also of research interest. To grind HFW by food waste disposer (FWD) at the source and dispose of it directly to the sewage treatment system could significantly reduce the amount of HFW disposed of in the MSW management system. Maalouf and El-Fadel [8] argued that using FWD can reduce greenhouse gas emissions by 42% and the overall cost by 17–28%. Edwards et al. [20] argued that FWD and anaerobic digestion scenarios would result in a similar environment and economic benefits. However, this is mainly attributed to adopting similar technical parameters for the anaerobic digestion treatment of sewage sludge and HFW due to the lack of inventory data on the former one. In a SLCC study [26], FWD had the highest budget costs due to the extremely high expense for the discharge process and no benefits from by-product revenue and GHG emission reductions. Yu and Li [61] compared the different HFW separation proportions and suggested that a HFW separation rate of 20% is acceptable where household time cost does not exceed the environmental cost. Mayer et al. [14] pointed out that thermal drying pre-treatment before HFW incineration would reduce the environmental and economic benefits. Carlsson et al. [58] found that physical pre-treatment (i.e., screw press) to divert more TS to the slurry could decrease GHG emissions and costs for an anaerobic digestion system.

Additionally, the collection and transportation process significantly affected the environment and economic values of the integrated HFW management system. Eriksson et al. [59] identified that the environmental burdens and economic costs between central sorting of mixed collected MSW and source separation of HFW are competitive. Elsewhere, Martinez-Sanchez et al. [25] and Slorach et al. [16] suggested that the individual collection and transportation process is predominantly responsible for the higher cost of anaerobic digestion treatment compared to co-incineration.

For the anaerobic digestion treatment system of HFW, the disposal procedures of digestate were crucial. Eriksson et al. [59] suggested that drying and pelleting of solid digestate is beneficial compared to direct spreading as fertilizer. Concerning wet digestate, dewatering and nitrogen utilization could decrease environmental impacts significantly but induce more economical costs in some cases compared to un-dewatering and spreading

on arable land. Mayer et al. [14] pointed out that the incineration of digestate demonstrated poor economic performance because the revenue of thermal recovery could not offset the cost of transportation and drying pre-treatment due to the lower LHV of digestate.

## 5. Conclusions and Recommendations

The use of environmental and economic life-cycle assessment is an essential auxiliary tool for carrying out HFW management decision making. This study reviewed related literature and drew the following conclusions and recommendations.

There are three main types of comprehensive environmental and economic assessment methodologies for HFW management systems: Societal LCC analysis, ECE analysis, and MCA. In practice, LCA for the environmental benefit analysis and LCC for the economic benefit analysis are derived from different discipline systems. The system boundaries, co-product allocation, and discounting approaches between LCC and LCA are not yet readily integrated. Most existing studies applied the ELCC method, a simplified ECE, which only makes qualitative comparisons between environmental and economic benefits. Some studies used the societal LCC method by converting environmental impacts into externality costs and incorporating them into the economic benefits. The quantitative conversion coefficients, i.e., monetary valuation parameters, are insufficiently studied. Therefore, in-depth research is needed on the coordination between LCA and LCC, the external valuation parameters of pollutant emissions in societal methods, and ECE standardization and evaluation methods.

According to the development tendency of policies and technologies, there are various HFW management strategies. Prevention is recommended as the most preferred option for FW; however, the discussion of this route has been rare up to now. In addition, limited research has been conducted on on-site HFW treatment, which has been presented in some Chinese cities in recent years. Thus, FW prevention and on-site HFW treatment technologies should be paid more attention when setting HFW management scenarios. In addition, the system boundaries for sorting, collection, transportation, product utilization, and residue disposal processes should be coherent with the investigated prevention and treatment technologies.

The inventory data in existing studies were primarily obtained from local survey reports and literature. In summary, data accumulation remains insufficient and a general inventory database has not yet been formed. On the one hand, individual studies are limited by the availability and applicability of inventory data. A typical example is that the types of environmental impacts investigated in the studies vary significantly. On the other hand, the study results are relatively independent, with weak continuity and inheritance of the entire research field. It is necessary to conduct long-term tracking of the HFW characteristics, treatment parameters, and material flow to accumulate inventory data uniquely for HFW management to provide reliable queries useful for researchers and decisionmakers.

**Author Contributions:** Conceptualization, N.Y. and B.Y.; Methodology, N.Y. and Y.L.; Validation, N.Y. and Z.D.; Formal Analysis, Y.L.; Investigation, F.L.; Resources, J.Z.; Data Curation, T.D. and Q.W.; Writing—Original Draft Preparation, N.Y., F.L., T.D. and Q.W.; Writing—Review and Editing, N.Y., Y.L. and B.Y.; Supervision, N.Y. and B.Y.; Project Administration, B.Y.; Funding Acquisition, N.Y. All authors have read and agreed to the published version of the manuscript.

**Funding:** This work was supported by the National Natural Science Foundation of China (Grant no. 51808349) and the Shenzhen Fundamental Research Program (Grant no. JCYJ20180302173924767).

**Institutional Review Board Statement:** Not applicable.

**Informed Consent Statement:** Not applicable.

**Conflicts of Interest:** The authors declare no conflict of interest.

**Abbreviations**

| | |
|---|---|
| ADP | Abiotic Depletion Potential |
| AHP | Analytical Hierarchy Procedure |
| AP | Acidification Potential |
| CASE | Cost Assessment for Sustainable Energy Systems |
| CBA | Cost–Benefit Analysis |
| CDM | Clean Development Mechanism |
| DEA | Data Envelopment Analysis |
| ECE | Environmental Cost Efficiency |
| EIA | Environmental Impact Assessment |
| ELCC | Environmental LCC |
| EP | Eutrophication Potential |
| FETP | Freshwater Eco-Toxicity Potential |
| FFDP | Fossil Fuel Depletion Potential |
| FU | Functional Unit |
| FW | Food Waste |
| FWD | Food Waste Disposer |
| GDP | Gross Domestic Product |
| GWP | Global Warming Potential |
| HFW | Household Food Waste |
| HKW | Household Kitchen Waste |
| HTP | Human Toxicity Potential |
| LCA | Life-Cycle Assessment |
| LCC | Life-Cycle Costing |
| LC-CBA | Life-Cycle Cost–Benefit Analysis |
| LCSA | Life-Cycle Sustainability Assessment |
| LHV | Lower Heating Value |
| MAET | Marine Aquatic Eco-Toxicity |
| MCA | Multicriteria Analysis |
| MSW | Municipal Solid Waste |
| ODP | Ozone-Layer Depletion Potential |
| OFMSW | Organic Fraction of Municipal Solid Waste |
| OW | Organic Waste |
| PM | Particulate Matter |
| POP | Photochemical Oxidation Potential |
| SFs | Sustainability Factors |
| SLCC | Societal LCC |
| SW | Solid Waste |
| TET | Terrestrial Eco-Toxicity |
| TOPSIS | Technique for Order Preference by Similarity to an Ideal Solution |
| WW | Wastewater |

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
