# Peer review of "Environmental and Economic Life-Cycle Assessments of Household Food Waste Management Systems: A Comparative Review of Methodology and Research Progress"

_sustainability, doi:10.3390/su14137533_

Round 1
Reviewer 1 Report
This manuscript presents a review on Household Food Waste Management Systems in terms of Environmental and Economic Life-cycle Assessments. The topic is interesting and timely. However, the writing of the manuscript should be improved substantially to make the content straightforward, clarity and easy understanding according to the comments below:
- The straightforwardness and story of the abstract is poor. The authors should improve it by highlighting the importance and issues of managing HFW with some quantitative data (e.g., annual production etc.), followed by precisely pointing out the objectives/key content of this review.
- In the introduction, the authors should briefly discuss how the environmental and Economic Life-cycle Assessments influence HFW management and compare relevant reviews, if possible, to extract or highlight the major gap(s) that this review will deal with.
- It would be nice if a figure can be included showing the major methods or aspects discussed in section 1 (Environmental and Economic Assessment Methods) (potential figure 1a) and Section 2 (Environmental and Economic Assessment of MSW Management Based on Life-cycle Theory) (potential figure 1b) including the key comparisons (methods/ advantages/Disadvantages or other important aspects).
- The contents of Table 1, 2 and 3 are too wordy and should be more precise.
- In section 3 (Research Progress of Environmental and Economic Life-cycle Assessment of HFW Management), the authors conducted a research work using secondary data. The aim is good. However, the results of such study are too descriptive and miss the appropriate presentation. In particular, Table 4 is too wordy and it is hard to get the key points.
Author Response
This manuscript presents a review on Household Food Waste Management Systems in terms of Environmental and Economic Life-cycle Assessments. The topic is interesting and timely. However, the writing of the manuscript should be improved substantially to make the content straightforward, clarity and easy understanding according to the comments below:
C1: The straightforwardness and story of the abstract is poor. The authors should improve it by highlighting the importance and issues of managing HFW with some quantitative data (e.g., annual production etc.), followed by precisely pointing out the objectives/key content of this review.
R1: The Abstract and Introduction have been revised to highlight the importance of HFW management. The quantitative data of annually FW production and the research objective of this paper have been clarified (Line 77-78, Page 5).
C2: In the introduction, the authors should briefly discuss how the environmental and Economic Life-cycle Assessments influence HFW management and compare relevant reviews, if possible, to extract or highlight the major gap(s) that this review will deal with.
R2: The Introduction has been revised as suggested (Line 112-135, Page 6).
C3: It would be nice if a figure can be included showing the major methods or aspects discussed in section 1 (Environmental and Economic Assessment Methods) (potential figure 1a) and Section 2 (Environmental and Economic Assessment of MSW Management Based on Life-cycle Theory) (potential figure 1b) including the key comparisons (methods/ advantages/Disadvantages or other important aspects).
R3: We have added Figure 1 to show the comparison among the methods in Section 1 and 2.
C4: The contents of Table 1, 2 and 3 are too wordy and should be more precise.
R4: The tables have been revised.
C5: In section 3 (Research Progress of Environmental and Economic Life-cycle Assessment of HFW Management), the authors conducted a research work using secondary data. The aim is good. However, the results of such study are too descriptive and miss the appropriate presentation. In particular, Table 4 is too wordy and it is hard to get the key points.
R5: The contents in Section 3 and Table 4 have been revised.

Reviewer 2 Report
The main problem is that I found many sentences suspected of plagiarism. The author is requested to reorganize the language, and it is also recommended that the editorial department check the duplicates. Other details are as follows:
- In Institution: China or People’s Republic of China, why contain two styles? This reflects the author's lack of rigours.
- In Abstract: Delete the second ‘comprehensively’ in Line 19.
- Line 20. Change to ‘three main’
- In 2.1 and 2.2 section. Too many ‘Someone presented…’, this mechanical superposition is a very low-level form of Review.
- In Table 1. Wrong word. ‘Defied’ change to ‘Defined’. And some other grammatical mistake, such as ‘Difference eco-efficiency’. And some wrong unit expression, such as ‘person yr-1’, not ‘person·yr’; Euro/t may change to Euro t-1. The unit forms in full text need unified expression.
- In 3.3. ‘Main Results’ is not a good subtitle. Please redesign the title.
- The content in the table is too complex, which is not conducive to typesetting. It is recommended that the author delete or reorganize it appropriately. Especially in Table 4.
- The Conclusion part is too long. It is recommended to simplify the conclusion, and other additional content can be used as an Outlook. So, the 4th section is Outlook, and the 5th Section is Conclusion.
Author Response
The main problem is that I found many sentences suspected of plagiarism. The author is requested to reorganize the language, and it is also recommended that the editorial department check the duplicates. Other details are as follows:
C1: In Institution: China or People’s Republic of China, why contain two styles? This reflects the author's lack of rigours.
R1: The country in institution have all been changed to China. (Line 6-8, Page 1)
C2: In Abstract: Delete the second ‘comprehensively’ in Line 19.
R2: The Abstract has been rewritten.
C3: Line 20. Change to ‘three main’
R3: The sentence has been revised as “This paper compares the three main environmental and economic assessment methodologies, i.e., societal life-cycle costing (Societal LCC), environmental cost-effectiveness (ECE) analysis, and multicriteria analysis (MCA) in terms of the definitions, method frameworks, and their advantages/disadvantages.” (Line 15-18, Page 2)
C4: In 2.1 and 2.2 section. Too many ‘Someone presented…’, this mechanical superposition is a very low-level form of Review.
R4: The descriptions in Section 2.1-2.3 have been revised. (Line 216-196, Page 10-13)
C5: In Table 1. Wrong word. ‘Defied’ change to ‘Defined’. And some other grammatical mistake, such as ‘Difference eco-efficiency’. And some wrong unit expression, such as ‘person yr-1’, not ‘person·yr’; Euro/t may change to Euro t-1. The unit forms in full text need unified expression.
R5: The grammatical mistake has been checked and revised.
According to Table 8 in Yang et al. (2015), the unit for normalized environmental and economic indicators is ‘person·yr’. For example, the economic cost of disposal for 1 ton of MSW are with the unit for CNY. To consider a standard unit in keeping with environmental impact potential, the economic cost were standardized using GDP per capita with the unit of CNY·person-1yr-1. Then, the normalized economic cost are with the unit of “person·yr”.
The other unit expressions in full text have been revised and unified.(see Table 1 and Table 2)
C6: In 3.3. ‘Main Results’ is not a good subtitle. Please redesign the title.
R6: The subtitle of section 3.3 has been changed to “Which management strategy is preferable?” (Line 415, Page 19)
C7: The content in the table is too complex, which is not conducive to typesetting. It is recommended that the author delete or reorganize it appropriately. Especially in Table 4.
R7: The tables have been revised.
C8: The Conclusion part is too long. It is recommended to simplify the conclusion, and other additional content can be used as an Outlook. So, the 4th section is Outlook, and the 5th Section is Conclusion.
R8: The title of Section 4 has been changed to “Conclusions and recommendations”. (Line 475, Page 26. Meanwhile, the context in Section 4 has been rewritten according to the main text.

Reviewer 3 Report
Review of “Environmental and Economic Life-cycle Assessments…”
There is some good content in the manuscript, but the writing and organization make it difficult for a reader to appreciate the content.
A couple of major issues: (1) the abstract concludes that AD is the most environmentally friendly technology, but none of the tables are designed to help the reader understand how that conclusion was derived, and (2) Table 4 is difficult to digest as it is so large.
Please see below for additional comments that could improve the manuscript.
The abstract is difficult to read. The second sentence is confusing as written (is the tradeoff between mitigating pollution and creating costs or is it between quality decision making and economic costs?). The terms ‘centralized treatment’ and ‘source sorting’ may not be understood by someone just reading the abstract. What other technologies beside AD were considered? How is ‘environmentally friendly’ defined? What is a ‘diverse’ conclusion?
Opening paragraph: why is there no detailed discussion of the GHG generation and impacts of HFW in MSW?
Line 64 – why does organic waste have 2 different acronyms: HKW and OW (line 75)? Same in Table 4. It would seem that HKW should stand for Household Kitchen Waste, which may or may not be entirely organic in content (e.g., some packaging waste).
Line 88. HFW is also the largest component of MSW in the United States, and perhaps in other developed countries as well.
Line 92. Contamination of recyclables is mentioned here, but is never discussed again in the paper.
Line 108 – “HFW is ground in the kitchen…” This makes it sound like this is how all HFW is handled in “…many cities in the United States…” No US city relies exclusively on this procedure – rather some portion of HFW in many cities may be handled this way. Perhaps “…in the United States, some portion of the HFW is ground…”
Line110 – what is meant by ‘squeeze’? Do you mean ‘de-water’?
Section 1 – it would be good to begin this section with a figure that simply visualizes the categories of assessments (terms only – no definitions) and the sub-types of each assessment that you explore in this paper. This would provide a simple hierarchical overview of the methods you will explore and help readers see a roadmap. Also, it would be useful for you to clarify if the LCA’s referred to in this manuscript are attributional LCA’s or consequential LCA’s. It would then be helpful at some point after all the assessment methods are described to provide an example in which all methods are deployed to assess the same hypothetical scenario (the simplest one possible to keep it brief) so that readers can appreciate the way the reviewed assessment approaches would differ if applied to the same question. This would be particularly useful as the HFW cases reviewed differ on both the assessment method used and the details of scenarios assessed, making it difficult for a reader to appreciate the differences across assessment methods.
Line 140 – the term ‘environmental friendliness’ is used in several places including here. Please use a more specific and precise term here and in all places where this term appears.
Line 234 – “…the MSW management system aims to decrease environmental pollution without creating economic benefits.” Some MSW systems create by products that aim to generate economic benefit – e.g., biogas.
Line 235 “In the previous ones…” It is difficult to understand which ‘previous ones’ you refer to – please clarify.
Tables 1 and 2 – please note in title if all ‘reviewed studies’ were MSW or HFW studies – tables should be able to be read in isolation without reference to the main body of text
377 – discounting – please comment on how environmental analysis deals with the long-time frame for methane emissions that come from placing organics into landfill. If a kg of HFW creates methane emissions occurring over, e.g., 10 years, are all 10 years of emissions simply summed up and counted as an instantaneous impact?
395 – ‘glance’ seems imprecise – isn’t this a ‘review’?
422 – it seems inconsistent to consider co-treatment with garden waste and sewage sludge but to not consider the impacts of food waste collected from food service and institutional settings. Wouldn’t a city need to choose the size of a facility based upon food waste collected at food service locations as well?
453 – ‘HKW’ – is this supposed to be HFW or OW?
3.3 – Main Results. Given that your main take away seems to be that AD is the technology that has the best assessment in the reviewed articles, I was expecting a Table or Figure that helped support this point. Table 4 is already too large to be very useful to most readers – I would recommend replacing some elements of Table 4 (e.g., ‘Approaches and ‘Cost Categories’) with the articles’ ranking of their own scenarios so that the pattern of AD performance can be visualized by the reader and the Table can be more readable?
The manuscript only analyzes different ways to treat Household Food Waste (HFW). Please let readers know why there is no analysis of Household Food Waste reduction and discuss this as a limitation of the work. Indeed, most would argue that reduction should be prioritized over treatment and, if reduction were effective, it would hold implications for sizing decisions for municipal systems of treatment. Also, please address, would any of the assessment methodologies be effective for evaluating competing HFW reduction interventions?
Author Response
There is some good content in the manuscript, but the writing and organization make it difficult for a reader to appreciate the content.
R1:A couple of major issues: (1) the abstract concludes that AD is the most environmentally friendly technology, but none of the tables are designed to help the reader understand how that conclusion was derived, and (2) Table 4 is difficult to digest as it is so large.
C1: The ranking of scenarios in the reviewed studies has been added in Table 4 to highlight the conclusion that AD is the environmental preferable technologies. The other information in Table 4 has also been clarified. Correspondingly, the Abstract and Section 3.3 have been rewritten according to the revised Table 4.
Please see below for additional comments that could improve the manuscript.
C2: The abstract is difficult to read. The second sentence is confusing as written (is the tradeoff between mitigating pollution and creating costs or is it between quality decision making and economic costs?). The terms ‘centralized treatment’ and ‘source sorting’ may not be understood by someone just reading the abstract. What other technologies beside AD were considered? How is ‘environmentally friendly’ defined? What is a ‘diverse’ conclusion?
Opening paragraph: why is there no detailed discussion of the GHG generation and impacts of HFW in MSW?
R2: The Abstract has been rewritten to solve those suggested problems.
The GHG generations of HFW in MSW has been added in the reviesd Paragraph 2 as follows “The high content of HFW would reduce the lower heating value (LHV) of MSW and subsequently the energy recovery efficiency of MSW incineration, which induces higher GHG emissions.” (Line 94-95, Page 5)
C3: Line 64 – why does organic waste have 2 different acronyms: HKW and OW (line 75)? Same in Table 4. It would seem that HKW should stand for Household Kitchen Waste, which may or may not be entirely organic in content (e.g., some packaging waste).
R3: The food waste generated from households is lack of consistent terminology, "household food waste", "food waste", "household kitchen waste", "organic waste", “organic fraction of municipal solid waste” could be found in existing studies. In Table 4, the terms for household food waste used in the literature are showed for consistency with the original studies.(see the table note a in Line 467, Page 24)
During the composition analysis for MSW, the packaging materials are usually separated from organic parts and are considered as papers, plastics or metals according to its texture.
C4: Line 88. HFW is also the largest component of MSW in the United States, and perhaps in other developed countries as well.
R4: Yes, HFW maybe the largest component of MSW in several developed countries. Nevertheless, the proportion of HFW in developing countries (35%-75%) are significantly higher than that in developed countries (20%-45%) according to our previous studies (Yang, 2018). To avoid ambiguous, the original sentence has been rewritten as “HFW is the main component of MSW, e.g., accounting for >50% of the total MSW in wet weight in developing countries.” (Line 91-92, Page 5)
Reference: Yang Na, Damgaard Anders, Scheutz Charlotte, Shao Li-Ming, He Pin-Jing, A comparison of chemical MSW compositional data between China and Denmark. Journal of Environmental Science. 2018, 74:1-10.
C5: Line 92. Contamination of recyclables is mentioned here, but is never discussed again in the paper.
R5: In paragraph 2, contamination of recyclables is described as one of the disadvantages for traditional treatment of HFW (i.e. mixed collected with residual waste). This is the background of this study. The purpose of this description is to emphasize the importance of source sorting and individual treatment of HFW, which is being occurred in Europe and several Chinese cities as mentioned in paragraph 3 and is the research objective of this paper. So the contamination of recyclables is not discussed in the following text.
C6: Line 108 – “HFW is ground in the kitchen…” This makes it sound like this is how all HFW is handled in “…many cities in the United States…” No US city relies exclusively on this procedure – rather some portion of HFW in many cities may be handled this way. Perhaps “…in the United States, some portion of the HFW is ground…”
R6: The original sentence has been revised as “In the United States, food waste disposers (FWDs) are introduced to some households, by which HFW is ground in the kitchen and discarded through the sewer for treatment with wastewater.” (Line 106-108, Page 5-6)
C7: Line110 – what is meant by ‘squeeze’? Do you mean ‘de-water’?
R7: The original sentence has been revised as “In some Chinese cities, HFW is pre-treated by high-pressure extrusion and delivered to incineration plant.” (Line 108-109 Page 6)
C8: Section 1 – it would be good to begin this section with a figure that simply visualizes the categories of assessments (terms only – no definitions) and the sub-types of each assessment that you explore in this paper. This would provide a simple hierarchical overview of the methods you will explore and help readers see a roadmap. Also, it would be useful for you to clarify if the LCA’s referred to in this manuscript are attributional LCA’s or consequential LCA’s. It would then be helpful at some point after all the assessment methods are described to provide an example in which all methods are deployed to assess the same hypothetical scenario (the simplest one possible to keep it brief) so that readers can appreciate the way the reviewed assessment approaches would differ if applied to the same question. This would be particularly useful as the HFW cases reviewed differ on both the assessment method used and the details of scenarios assessed, making it difficult for a reader to appreciate the differences across assessment methods.
R8: Figure 1 is created to show the comparison among the methods in section 1 and 2.
C9: Line 140 – the term ‘environmental friendliness’ is used in several places including here. Please use a more specific and precise term here and in all places where this term appears.
R9: The sentences using the term “environmental friendliness” have been revised as follows:
“Anaerobic digestion is environmentally preferable to composting and landfilling; it is comparable to biodiesel production, feeding conversation, and incineration.” (Line 24-26, Page 2)
“…, what policy or project is better to achieve lower environmental impacts and affordable economic costs?” (Line 139-140, Page 7)
“From the perspective of environmental benefit, anaerobic digestion is preferable to composting and landfilling and is comparable with biodiesel production, feeding conversation, and incineration.” (Line 419-421, Page 19)
C10: Line 234 – “…the MSW management system aims to decrease environmental pollution without creating economic benefits.” Some MSW systems create by products that aim to generate economic benefit – e.g., biogas.
R10: This sentence aims to emphasize the difference between MSW management system and the industrial processes. In some MSW system, economic benefit are generated from by-products sales, but this is not the primary objective. To avoid ambiguous, the original sentence hans been revised as “The primary objective of the MSW management system is to decrease environmental pollution rather than create economic benefits.” (Line 208-209, Page 9)
C11: Line 235 “In the previous ones…” It is difficult to understand which ‘previous ones’ you refer to – please clarify.
R11: The original sentence has been revised as “During the environmental and economic assessment of construction and industrial processes, the current costs and benefits are usually considered. In contrast, the assessment approach for the MSW management system considers the environmental impacts of its whole life-cycle in addition.” (Line 209-212, Page 9-10)
C12: Tables 1 and 2 – please note in title if all ‘reviewed studies’ were MSW or HFW studies – tables should be able to be read in isolation without reference to the main body of text
R12: The reviewed studies mentioned in Table 1 and Table 2 referred to MSW. To clarify, the titles have been revised as “Table 1. Approaches used for environmental cost efficiency in reviewed studies for the MSW management system” and “Table 2. Approaches used for multicriteria analysis method in reviewed studies for the MSW management system”.
C13: Line 377 – discounting – please comment on how environmental analysis deals with the long-time frame for methane emissions that come from placing organics into landfill. If a kg of HFW creates methane emissions occurring over, e.g., 10 years, are all 10 years of emissions simply summed up and counted as an instantaneous impact?
R13: Environmental analysis does not need discounting, considering that the pollutant emissions occured in different time induce the same impact to the environment. Taking LCA of MSW landfiiling as an example (Yang, et al. 2013), the GHG emissions during the whole life time are calculated year by year and sumed up as total emission value per tonne of HFW with no discounting among time. The explanation for discounting of environmental analysis has been added in the revised manuscript as follows: “Generally, the environmental analysis does not need discounting, considering that the pollutant emissions occurred at different times induce the same impact on the environment.”(Line 327-328, Page 15).
Reference: Yang Na, Zhang Hua, Shao Li-Ming, Lü Fan, He Pin-Jing, Greenhouse gas emissions during MSW landfilling in China: Influence of waste characteristics and LFG treatment measures. Journal of Environmental Management 2013, 129: 510-521.
C14: Line 395 – ‘glance’ seems imprecise – isn’t this a ‘review’?
R14: “glance” has been revised as “systematic review” (Line 341, Page 17).
C15: Line 422 – it seems inconsistent to consider co-treatment with garden waste and sewage sludge but to not consider the impacts of food waste collected from food service and institutional settings. Wouldn’t a city need to choose the size of a facility based upon food waste collected at food service locations as well?
R15: In this paper, HFW refers to the food waste generated from household rather than from restaurants, markets and the food industry (see revised paragraph 1 in Introduction). So, the scenarios for food waste collection at food service and institutional settings are not the research objective of this study.
C16: Line 453 – ‘HKW’ – is this supposed to be HFW or OW?
R16: The terms “HKW” in section 3 have been changed to “HFW” (Line 387, Page 18, Line 429, 439, Page 20)
C17: 3.3 – Main Results. Given that your main take away seems to be that AD is the technology that has the best assessment in the reviewed articles, I was expecting a Table or Figure that helped support this point. Table 4 is already too large to be very useful to most readers – I would recommend replacing some elements of Table 4 (e.g., ‘Approaches and ‘Cost Categories’) with the articles’ ranking of their own scenarios so that the pattern of AD performance can be visualized by the reader and the Table can be more readable?
R17: Table 4 has been revised as recommended. The key information, such as system boundary, scenario ranking and data resources, are clarified to increase the readability of Table 4. Correspondingly, Section 3.3 (line 416-428, Page 19-20) has been rewritten according to revised Table 4.
C18: The manuscript only analyzes different ways to treat Household Food Waste (HFW). Please let readers know why there is no analysis of Household Food Waste reduction and discuss this as a limitation of the work. Indeed, most would argue that reduction should be prioritized over treatment and, if reduction were effective, it would hold implications for sizing decisions for municipal systems of treatment. Also, please address, would any of the assessment methodologies be effective for evaluating competing HFW reduction interventions?
R18: The discussion of HFW prevention has been added in Section 3.3 as follows “Consistent with the FW management hierarchies, prevention of edible FW is demonstrated as the best route over treatment or utilization. This is due to the avoided food production process, which contributes to extremely high environmental and economic impacts in the scenarios without FW prevention.” (Line 416-419, Page 19) and Section 4 as follows “Prevention is recommended as the most preferred option for FW, whereas discussion of this route is rarely up to now. … Thus, FW prevention and on-site HFW treatment technologies should be paid more attention to when setting HFW management scenarios.”(Line 492-495, Page 26).

Round 2
Reviewer 1 Report
The authors have revised the manuscript according to the comments. So, I recommend acceptance of the revised version of the manuscript.
Reviewer 2 Report
no further comments